# A Theoretical Investigation into the Oligomer Structure of Carbon Dots Formed from Small-Molecule Precursors

**DOI:** 10.3390/molecules29122920

**Published:** 2024-06-19

**Authors:** Chunlan Li, Xu Zhu, Maotian Xu

**Affiliations:** Henan Key Laboratory of Biomolecular Recognition and Sensing, Henan Joint International Research Laboratory of Chemo/Biosensing and Early Diagnosis of Major Diseases, College of Chemistry and Chemical Engineering, Shangqiu Normal University, Shangqiu 476000, China; chunlan16@yeah.net (C.L.); xumaotian@sqnu.edu.cn (M.X.)

**Keywords:** carbon dots, oligomers, small-molecule precursors, molecule structure, density functional theory

## Abstract

In-depth insights into the oligomers of carbon dots (CDs) prepared from small-molecule precursors are important in the study of the carbonization mechanism of CDs and for our knowledge of their complex structure. Herein, citric acid (CA) and ethylenediamine (EDA) were used as small-molecule precursors to prepare CDs in an aqueous solution. The structure of oligomers acquired from CA and EDA in different molar ratios and their formation process were first studied using density functional theory, including the dispersion correction (DFT-D3) method. The results showed that the energy barrier of dimer cyclization was higher than that of its linear polymerization, but the free energy of the cyclized product was much lower than that of its reactant, and IPCA (5-oxo-1,-2,3,5-tetrahydroimidazo [1,2-a]pyridine-7-carboxylic acid) could therefore be obtained under certain conditions. The oligomers obtained from different molar ratios of EDA and CA were molecular clusters formed by short polyamide chains through intermolecular forces; with the exception of when the molar ratio of EDA to CA was 0.5, excessive CA did not undergo an amidation reaction but rather attained molecular clusters directly through intermolecular forces. These oligomers exhibited significant differences in their surface functional groups, which would affect the carbonization process and the surface structure of CDs.

## 1. Introduction

Carbon dots (CDs), especially doped CDs, have been found to be great promising materials for applications in optoelectronics, photocatalysis, bioimaging, biosensing, drug delivery, and therapy due to their advantages of photoluminescence, edge effects, excellent water solubility, low cytotoxicity, and good biocompatibility [1,2,3,4,5,6,7]. Presently, “top-down” and “bottom-up” methods are commonly used to prepare CDs. The “top-down” method mainly involves cutting large-sized carbon precursors (such as graphite, graphene, carbon nanotubes, carbon fibers, and carbon black) into small-sized carbon quantum dots using physical or chemical methods. The “bottom-up” methods involve using small molecules as precursors to obtain larger carbon quantum dots through a series of chemical reactions [7]. Compared with the “top-down” method, the “bottom-up” method is generally considered more suitable for the preparation of doped CDs due to its ability to regulate molecule precursors and reaction processes to control the chemical composition and structure of the product [7,8,9]. However, the process of preparing doped CDs from small-molecule precursors using the “bottom-up” method is very complex, and in-depth insights into the carbonization mechanism of doped CDs and their complex structure still present a great challenge. It is well known that the structure of a material determines its performance, and performance determines its function and application. The structure of doped CDs is still unclear, and the relationship between their structure and performance cannot be established, which leads to limitations in many applications.

In general, the synthesis of doped CDs using the “bottom-up” method requires that small-molecule precursors have active functional groups or unsaturated additive bonds [10,11], such as carboxyl-, hydroxyl-, and nitrogen-related groups. Among these groups, nitrogen-related groups have great impacts on the spectral properties of CDs [12,13], and much research has been devoted to the preparation of nitrogen-doped CDs and the study of their structure and properties [14,15,16,17,18]. Strauss et al. [14] prepared nitrogen-doped CDs using citric acid (CA) as a carbon source and urea as a nitrogen-doping source. Through the analysis of absorption spectra, fluorescence spectra, Fourier-transform infrared (FTIR), and nuclear magnetic resonance (NMR) spectra, they proposed a possible reaction mechanism between CA and urea. Song et al. [15] prepared nitrogen-doped CDs from CA and ethylenediamine (EDA) and discussed the chemical structure of the obtained CDs to study their photoluminescence mechanism. They separated the obtained CDs by column chromatography on silica and obtained three types of products, including CDs with a carbon core, small-molecule IPCA (5-oxo-1,-2,3,5-tetrahydroimidazo [1,2-a]pyridine-7-carboxylic acid), and some oligomers. The precise structure of IPCA was proved by FTIR, NMR, and liquid chromatography–mass spectrometry (LC-MS) techniques, but the structure of oligomers and CDs with a carbon core was not clear. Vallan et al. [16] characterized the structure of CDs and proposed the structure of oligomers, including the dimer and octamer. They considered that these oligomers were obtained through a simple amide chain reaction of CA and EDA. Obviously, the formation of oligomers from small-molecule precursors is the first essential elementary stage in obtaining doped CDs and is also a key part of the whole preparation process. The structure of simple compounds can be proven through conventional analytical techniques, such as FTIR, NMR, and X-ray photoelectron spectroscopy (XPS). However, due to the difficulty in directly detecting intermediates obtained during the one-step hydrothermal method for preparing CDs in real time, the oligomer structure of CDs is difficult to analyze using conventional analytical techniques [15,19]. DFT calculation, as a commonly used theoretical method to study the structure or reaction mechanism of various compounds [20,21,22,23], can overcome this shortcoming.

In this work, CA and EDA were selected as classical small-molecule precursors to theoretically study the oligomers of nitrogen-doped CDs, while water was chosen as a solvent because it was cheap and environmentally friendly, making it a good choice for the experimental preparation of CDs. The electrostatic potential (ESP) of CA was evaluated with Multiwfn to analyze its surface reaction sites. The reaction process of CA and EDA in different molar ratios to obtain oligomers was investigated in depth using the DFT method for the first time. Based on these results, it is expected that we will be able to provide essential theoretical guidance for controlling the experimental conditions and knowing the structure of the final CDs, thus further improving the performance of the resulting CDs and meeting their tailored application requirements.

## 2. Results and Discussion

### 2.1. Molecular Geometry

In order to study the structural evolution of oligomers obtained by the reaction of CA and EDA, the molecule configurations of these reactants in an aqueous solution were first optimized, respectively, as shown in Figure 1A. The corresponding lowest frequency and Gibbs free energy of CA and EDA are presented in Appendix A. Clearly, neither of these structures has an imaginary vibrational frequency, indicating that the structure of CA and EDA is a local minimum point.

From Figure 1A, EDA contains two structurally symmetrical amino groups, both located at the end and structurally symmetrical, exhibiting similar chemical environments. CA contains an asymmetrical distribution of three carboxylic acids and one hydroxyl group, which may affect its reaction process. On this account, the ESP diagram of CA was investigated to predict its overall reactivity on the molecular surface, as demonstrated in Figure 1B. As can be seen, the surface minimum value of the ESP (−56.81 kcal∙mol^−1^) of CA appears near the carboxyl group containing the C9 atom, indicating that the carboxyl group containing the C9 has the most electrons clustered around it and is the main reaction site. Next are the carboxyl group containing the C18 atom (−43.54 kcal∙mol^−1^) and the carboxyl group containing the C3 atom (−37.40 kcal∙mol^−1^). The surface minimum value of the ESP for the hydroxyl group of CA (−10.84 kcal∙mol^−1^) is 45.97 kcal∙mol^−1^ higher than that of the carboxyl group containing the C9 atom, implying that the hydroxyl group of CA does not participate easily in the subsequent reaction.

### 2.2. Dimers

In view of the possible reaction of carboxyl groups with hydroxyl and amino groups during the preparation of CDs [16,17,24,25,26], the dimerization process of two CA molecules and the formation process of dimers through an amidation reaction between CA and EDA were studied, respectively.

Since the Gibbs free energy of the product (di-CA-IM3 + H_2_O) is 48.0 kJ∙mol^−1^ higher than that of the reactants (Appendix A), the dimerization of two CA molecules through the hydroxyl group is very challenging, which is in agreement with the above ESP result. Figure 2 displays the dimerization process of EDA with three different carboxyl groups of CA. The first path (path 1) is the reaction of EDA and the carboxyl group containing the C9 atom. The second path (path 2) is the reaction between EDA and the carboxyl group containing the C18 atom. The third path (path 3) is the reaction of EDA and the carboxyl group containing the C3 atom. As can be seen, the energy barrier of path 1 is relatively the lowest (177.7 kJ∙mol^−1^), followed by path 2 (187.6 kJ∙mol^−1^), indicating that the carboxyl group containing the C9 atom in CA is the main reaction site; this is also consistent with the above ESP result. In other words, the amino group (H23–N22–H24) of EDA will preferentially approach the carboxyl group containing the C9 atom in CA rather than the carboxyl group containing C18 and C3. Then, the N22 atom of EDA connects with the C9 atom in CA to gain a new bond, C9–N22 (1.53 Å), but the bond length of C9–O13 increases from 1.35 Å to 2.18 Å. Meanwhile, the H24 atom shifts from N22 to O13 via transition state di-TS1-1, leading to the formation of the intermediate di-IM2-1. Finally, the water molecule (H14–O13–H24) leaves the intermediate di-IM2-1 to produce the final dimerization product di-IM3-1, with free energy decreased by 27.0 kJ∙mol^−1^, which is energetically favorable.

However, molecules may also obtain dimers through weak interactions, such as hydrogen bonds and electrostatic forces, which can be calculated using the dispersion-corrected DFT-D3 method to be closer to the actual results [27,28]. The lowest frequency and Gibbs free energy of dimeric isomers formed under the weak interaction of CA and EDA are shown in Appendix A. Among ten dimeric isomers, the Gibbs free energy of di-IM1-1-1 is the lowest, indicating that the dimer di-IM1-1-1 developed by a weak interaction is the one with the most stable structure. However, compared with the product di-IM3-1 + H_2_O, the Gibbs free energy of the dimer di-IM1-1-1 is 15.0 kJ∙mol^−1^ higher than that of the product di-IM3-1 + H_2_O. This result means that CA and EDA in an aqueous solution are more likely to produce the product di-IM3-1 through an amidation reaction.

Figure 3 presents the cyclization process of di-IM3-1 in an aqueous solution. It can be seen that the N22 atom in di-IM3-1 first approaches the C18 atom, while the H23 atom leaves N22 and transfers to the O20 atom of the carboxyl group via the transition state Huan-TS1-1 to attain the intermediate Huan-IM2-1. Then, the water molecule (H23–O20–H21) leaves the intermediate to obtain the product Huan-IM3-1, containing a five-membered ring, with a potential energy reduction of 35.7 kJ∙mol^−1^. Next, another N31 atom closes to the C18 atom, while its connected H32 atom migrates to the O19 atom to gain a hydroxyl group (O19–H32) through the transition state Huan-TS2-1 to produce the product Huan-IM5-1 with two five-membered rings.

The Huan-IM5-1 produced has high energy, and its structure is unstable. Two hydroxyl groups of Huan-IM5-1 dehydrate easily with hydrogen atoms in the adjacent ethyl group to form double bonds, and finally, Huan-IM11-1 with a conjugated structure is obtained. Huan-IM11-1’s conjugated double bonds are not in the same ring, and its potential energy is 50.7 kJ∙mol^−1^ lower than that of di-IM3-1.

Figure 4 shows the first cyclization path of di-IM3-2 in an aqueous solution. It can be seen that the carboxyl carbon atom attacked by the N22 atom is C3. Then, the H23 atom bonded with N22 shifts to the O1 atom via the first transition state Huan-TS1-2 to produce the intermediate Huan-IM2-2. Finally, the first product Huan-IM3-2 is obtained. Since the carboxyl group containing C9 is directly connected to the central C8 atom, while the carboxyl group containing C18 and C3 is connected to C8 through the ethyl group (–CH_2_), a six-membered ring is obtained after the carboxyl group, which contains C3 attacked by N22. Next, the N31 atom of the terminal amino group continues to attack C3, while the H32 atom transfers to O4 to generate a hydroxyl group. The second Huan-IM5-2 product is formed, containing one six-membered ring, and one five-membered ring (two rings sharing the same side) is formed. Finally, the hydroxyl group in Huan-IM5-2 dehydrates with the hydrogen atom of the adjacent ethyl group to produce a double bond, and the final Huan-IM11-2 product is obtained. Huan-IM11-2’s potential energy is 149.0 kJ∙mol^−1^ lower than that of the reactant di-IM3-2. Compared with the initial reactant (CA + EDA), the free energy of Huan-IM11-2 decreases by 178.1 kJ∙mol^−1^, while the free energy of Huan-IM11-1 only decreases by 111.7 kJ∙mol^−1^, indicating that the generation of Huan-IM11-2 possesses obvious thermodynamic advantages.

The structure of Huan-IM11-2 is consistent with that of IPCA (5-oxo-1,-2,3,5-tetrahydroimidazo [1,2-a]pyridine-7-carboxylic acid) reported in the literature [15]. The two double bonds (C18=C15, C8=C5) obtained through the dehydration of Huan-IM5-2 are in the same ring, forming conjugated double bonds. This is the main reason for the significant reduction in the thermodynamic energy of the final cyclization product (Huan-IM11-2) obtained after the cyclization and dehydration of di-IM3-2.

For di-IM3-2, the N22 atom can also attack the C9 atom to produce a five-membered ring structure during its cyclization, as depicted in Appendix A. Compared with the first cyclization path (Figure 4), the cyclization energy barrier of the second path (the first reaction step) is lower than 18.6 kJ∙mol^−1^, but the energy barrier required for the dehydrogenation of the cyclized product to obtain conjugated double bonds at the second cyclization path (133.2 kJ∙mol^−1^ and 241.4 kJ∙mol^−1^, respectively) is much higher than that of the first cyclization path (93.4 kJ∙mol^−1^ and 97.1 kJ∙mol^−1^, respectively). Clearly, di-IM3-2 preferentially produces the cyclization product IPCA through the first cyclization path.

As mentioned above, the energy barrier for producing di-IM3-2 from CA and EDA is 9.9 kJ∙mol^−1^ higher than that for producing di-IM3-1, with the result that the main dimer products are the di-IM3-1 molecules. However, due to the structural advantages and thermodynamic energy advantages of the final Huan-IM11-2 (IPCA) product, the reaction between CA and EDA under certain conditions will also produce some di-IM-3-2 molecules for the obtention of IPCA.

### 2.3. Oligomers Formed by Equal Molar Ratios of EDA and CA

Figure 5 shows the linear polymerization process of two di-IM3-1 molecules in an aqueous solution. As can be seen, this polymerization mechanism is similar to the above mentioned dimerization mechanism of CA and EDA. First, the N58 atom of the amino group in one di-IM3-1 molecule close to the C18 atom of the carboxyl in another di-IM3-1 molecule forms the intermediate te-IM4-1. Then, the N58 atom connects with C18 to engender a new bond (N58–C18, 1.58 Å), but the bond length of C18–O20 increases to 2.14 Å. At the same time, the H59 atom transfers to O20 via the transition state to acquire the intermediate te-IM5-1. Finally, a water molecule (H21–O20–H59) leaves the intermediate to produce the tetramer product te-IM6-1, with a decrease in energy by 31.2 kJ∙mol^−1^. During the whole reaction process, the energy barrier of di-IM3-1 for linear polymerization is 150.2 kJ∙mol^−1^, which is lower than that of its first cyclization step (232.9 kJ∙mol^−1^) by 82.7 kJ∙mol^−1^. This result implies that di-IM3-1 is more prone to linear polymerization to obtain the tetramer te-IM6-1.

However, the oligomeric isomer te-IM4-1-1 attained under the weak interaction of two di-IM3-1 molecules has the lowest free energy (Table 1), and its Gibbs free energy is 9.3 kJ∙mol^−1^ lower than that of the product te-IM6-1 + H_2_O. This result suggests that two di-IM3-1 molecules are more likely to spontaneously form te-IM4-1-1 through an intermolecular interaction rather than forming te-IM6-1 through an amidation reaction. Since the free energy of te-IM4-1-4 is only 2.3 kJ∙mol^−1^ higher than that of te-IM4-1-1, the oligomer products obtained by two di-IM3-1 molecules include te-IM4-1-4 and te-IM4-1-1.

Figure 6A displays the optimized structure of te-IM4-1-1 and te-IM4-1-4. They contain several hydrogen bonds generated by carboxyl groups and amino groups. Protons in the carboxyl groups can migrate between molecules to produce carboxylate anions and amino cations. This may be related to the number of carboxyl and amino groups in the molecule. When the molar ratio of EDA to CA is the same, the number of carboxyl groups is more than that of the amino groups, and the solution is acidic, which makes it easy for the protons of carboxyl groups in the molecule to ionize and migrate. This phenomenon indicates that the oligomers obtained by an equal molar ratio of EDA and CA have good water solubility.

### 2.4. Oligomers Formed by Unequal Molar Ratios of EDA and CA

#### 2.4.1. The Molar Ratio of EDA to CA Is 2.0

Although di-IM3-1 molecules are readily formed into tetramers through intermolecular hydrogen bonds, they tend to polymerize with small molecules of EDA to yield the product tri-IM6. This is due to the presence of excessive EDA when the molar ratio of EDA to CA is 2.0, as depicted in Figure 7. Similar to the dimerization mechanism of CA and EDA, the amino group containing N38 undergoes an amidation reaction with the carboxyl group containing C18 to produce tri-IM6. The energy barrier of this process is 119.7 kJ∙mol^−1^. The free energy of tri-IM6 is 43.7 kJ∙mol^−1^ lower than that of its reactants, indicating that this polymerization process is energetically favorable. From the Gibbs free energy of ten isomers generated under the weak interaction of di-IM3-1 and EDA (Appendix A), Tri-IM4-1 has the lowest free energy, but its energy is 11.7 kJ∙mol^−1^ higher than the product tri-IM6 + H_2_O. This result indicates that the products obtained from di-IM3-1 and EDA are tri-IM6 + H_2_O.

Since tri-IM6 contains one carboxyl and one active amino group, it can continue to polymerize with itself, as presented in Figure 8. Although this reaction process is energetically favorable, the free energy of the intermediate hex-IM7 is lower than that of the product hex-IM9 + H_2_O, implying that the structure of the intermediate hex-IM7 is the most stable among all products. In addition, the free energy of hex-IM7 is also lower than that of its isomers acquired at the same calculation level (Appendix A), indicating that two tri-IM6 molecules are more easily connected by intermolecular hydrogen bonds than by the amidation reaction.

It is clear that when the molar ratio of EDA to CA is 2.0, the main oligomer product obtained is hex-IM7, whose structure is depicted in Figure 6B. This oligomer contains four active amino groups and two carboxyl groups. Similar to te-IM4-1-1 (or te-IM4-1-4), hex-IM7 is also composed of hydrogen bonds. Due to the fact that the number of amino groups is higher than that of carboxyl groups, the proton in the carboxyl group of hex-IM7 is not ionized and migrated so that there are no carboxyl anions and amino cations in its structure.

#### 2.4.2. The Molar Ratio of EDA to CA Is 3.0 

When the molar ratio of EDA to CA is 3.0, three EDA molecules will react with one CA molecule. That is, on the basis of the reaction between two EDA molecules and CA, the third EDA molecule will continue to participate in the subsequent reaction, as described in Figure 9. The energy barrier of this process is 193.1 kJ∙mol^−1^. The final product CA-3EDA-IM3 + H_2_O has the lowest free energy and is 42.3 kJ∙mol^−1^ lower than the reactants, making it energetically favorable. Since the free energy of the isomers obtained from intermolecular interactions was always higher than that of CA-3EDA-IM3 + H_2_O (Appendix A), the main product obtained from tri-IM6 and EDA was CA-3EDA-IM3 + H_2_O.

Table 2 lists the Gibbs free energy of CA-3EDA-IM3 and its oligomeric isomers gained under weak interactions in an aqueous solution. Compared to CA-3EDA-IM3 + CA-3EDA-IM3, the free energy of oct-IM1-1, oct-IM1-2, oct-IM1-4, and oct-IM1-10 are decreased, with oct-IM1-1 having the largest decrease (13.8 kJ∙mol^−1^) among these isomers, followed by oct-IM1-2, whose free energy was 2.1 kJ∙mol^−1^ higher than oct-IM1-1. These results indicate that the product CA-3EDA-IM3 can be combined through intermolecular interactions to produce oct-IM1-1 and a few oct-IM1-2 products. Both oct-IM1-1 and oct-IM1-2 contain six active amino groups but do not have carboxyl groups, as shown in Figure 6C. Obviously, the number of amino groups is higher than that of carboxyl groups at this molar ratio; protons in the carboxyl groups of oct-IM1-1 and oct-IM1-2 will not ionize and migrate, meaning that their structure is free of carboxyl anions and amino cations.

#### 2.4.3. The Molar Ratio of EDA to CA Is 0.5

When CA is excessive and the molar ratio of EDA to CA is 0.5, two CA molecules will react with one EDA molecule. In other words, the dimer di-IM3-1 obtained from CA and EDA will continue to react with another CA molecule, as shown in Figure 10. The reaction mechanism of di-IM3-1 and CA is similar to that described previously, and the free energy of the product 2CA-EDA-IM3 is the lowest in the whole process. However, the free energy of the isomer (2CA-EDA-IM1-1) acquired under the weak interaction of di-IM3-1 and CA has the lowest free energy and is 30.8 kJ∙mol^−1^ lower than that of the product 2CA-EDA-IM3 + H_2_O (Table 3), implying that CA will preferentially form the trimer 2CA-EDA-IM1-1 with di-IM3-1 through an intermolecular interaction rather than undergoing an amidation reaction.

Figure 6D displays the structure of the trimer 2CA-EDA-IM1-1. Since the number of carboxyl groups is far higher than that of amino groups at this molar ratio, protons on the carboxyl and hydroxyl groups will migrate simultaneously, making the amino group –NH_3_^+^. This trimer, containing four carboxyl groups, one –NH_3_^+^ ion, and one –COO^−^ ion, is (relatively) the most stable among all isomers and has good water solubility, which is the main oligomer product at this molar ratio.

As mentioned above, excessive EDA and excessive CA have different reaction characteristics during the generation of oligomers in aqueous solutions. Excessive EDA preferentially participates in the amidation reaction to gain oligomers and then produce clusters through intermolecular forces; excessive CA does not participate in the amidation reaction but directly forms clusters through intermolecular forces. The amidation reaction leads to the disappearance of original functional groups, while clusters retain the original functional groups on the surface of the product. The functional groups exposed on the surface of clusters may be retained and delivered to the surface of the final CDs, which can provide theoretical guidance for experimental research on the modulation and functionalization of surface functional groups in CDs. Moreover, knowing the types and distribution of functional groups in clusters will also help in the study of the carbonization mechanism of CDs. 

## 3. Computational Methods

DFT methods were performed using the Gaussian 09 software package [29], which employs a 6-311+G** basis set, utilizes the B3LYP as the functional component, and includes D3 correction to consider dispersion interactions. The geometric configuration optimization and frequency calculation of all species in this work were carried out under the consideration of the conductor-like polarizable continuum model (CPCM, water, ε = 78.4). In the preliminary study of the reaction process of small-molecule compounds in aqueous solutions, CPCM approximation could be used to consider the impact of bulk water [30,31,32,33]. The vibration frequencies were calculated while optimizing the structure of reactants or products and calculating the transition state. The integral=ultrafinegrid keyword was used simultaneously to ensure the reliability of the obtained frequencies. The calculated vibration frequencies could be used to judge whether the optimized structure was a local minimum point (zero imaginary frequencies) and whether there was only one imaginary frequency in the transition state. Meanwhile, the intrinsic reaction coordinate (IRC) calculations [34,35] were conducted to verify the transition state. The Gibbs free energy was obtained from the addition of the zero-point energy correction (E_ZPE_, 0 K), the thermal correction (E_corr_, 298 K), the entropy correction (-TS_298_), and the electron energy (E_elect_, 0 K). The thermal correction and the entropy correction were acquired through analytical thermochemistry at 1 atm and 298.15 K. Zero-point energy corrections were performed based on the vibration frequency calculations at the same level.

The ESP of the CA molecule was analyzed and mapped onto the isodensity surface by combining Multiwfn [36] with the VMD 1.9.3 program [37] based on the wavefunction obtained from the Gaussian 09 output file. The Molclus program [38] was utilized to find the most stable configuration of the oligomers obtained by molecular clusters. First, 100 initial configurations were randomly received by the genmer program under Molclus 1.9 software, and the MOPAC2016 program [39] was used for preliminary optimization at the PM6-DH+ level through a semi-empirical method. Then, ten low-lying configurations were selected and optimized at the dispersion-corrected B3LYP-D3/6-311+G** level, and the configuration with the lowest free energy was obtained.

## 4. Conclusions

In summary, the oligomers of CDs at different molar ratios of EDA and CA were obtained using the DFT-D3 method. The formation mechanisms of small-molecule IPCA were proven from an energetic perspective. The carboxyl group containing C9 in CA was the main reaction site, and it preferentially reacted with the amino group of EDA to obtain the dimer di-IM3-1. However, the dimer di-IM3-2 could also be obtained and then cyclized to produce Huan-IM11-2 (IPCA) with a larger energy advantage. The oligomers attained from the linear polymerization of di-IM-3-1 when the molar ratios of EDA to CA were 1.0, 2.0 and 3.0, including te-IM4-1-1 and te-IM4-1-4, hex-IM7, oct-IM1-1 and oct-IM1-2, were all initially formed through an amidation reaction to gain short polyamide chains and then corresponding short polyamide chain spontaneous clustering. However, when the molar ratio of EDA to CA was 0.5, excessive CA did not undergo an amidation reaction with a primary amino group but directly generated clusters with the dimer di-IM3-1 by intermolecular forces. These specific reaction traits of EDA and CA directly influenced the oligomer structure, leading to oligomers with a molar ratio of 2.0 and 3.0 exhibiting an increased presence of primary amino groups. Conversely, oligomers produced with a molar ratio of 0.5 and 1.0 exhibited a higher concentration of carboxyl groups along with carboxyl anions and amino cations, thereby enhancing their water solubility.

The structure characteristics of these oligomers will have great influence on the carbonization process of CDs and their surface structure, which will be further explored in our later work. The methodology utilized in this work can also be used to investigate the structure of oligomers obtained from other types of precursors with different functional groups to guide the corresponding experiments and explore the complex structure of CDs, which is very important from the perspectives of foundation and application.

## Figures and Tables

**Figure 1 molecules-29-02920-f001:**
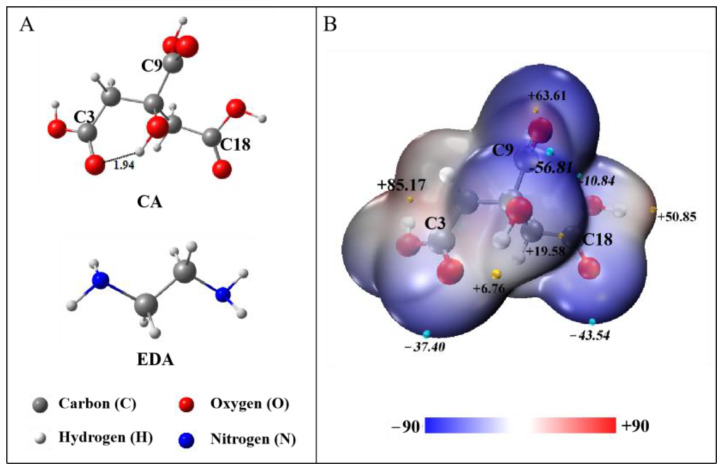
(**A**) The optimized geometries of CA and EDA in aqueous solution, with carbon in dark grey, hydrogen in light grey, oxygen in red, and nitrogen in blue; (**B**) the electrostatic potential (ESP) mapped onto the isodensity surface (0.001 a.u.) of CA. The unit is in kcal∙mol^−1^. The surface local maxima and minima of ESP are expressed as orange and cyan spheres, respectively.

**Figure 2 molecules-29-02920-f002:**
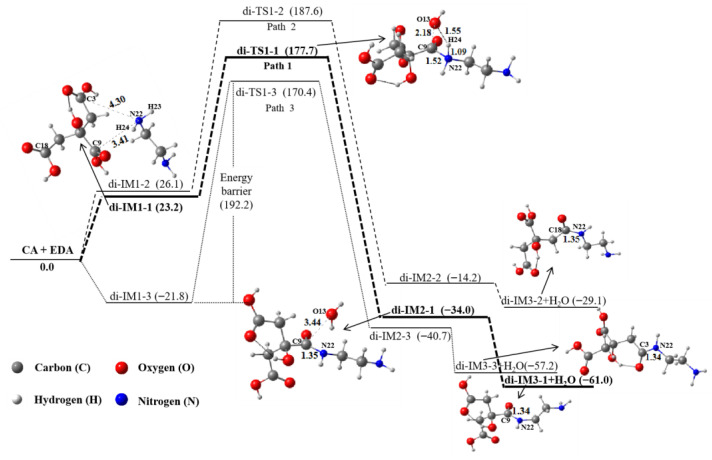
The dimerization of EDA and CA in an aqueous solution. The relative energies are in kJ∙mol^−1^. This unit is uniformly used in all figures.

**Figure 3 molecules-29-02920-f003:**
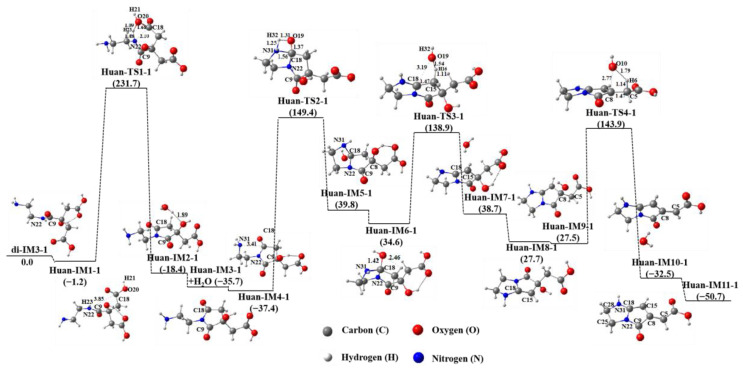
Cyclization process of di-IM3-1 in aqueous solution.

**Figure 4 molecules-29-02920-f004:**
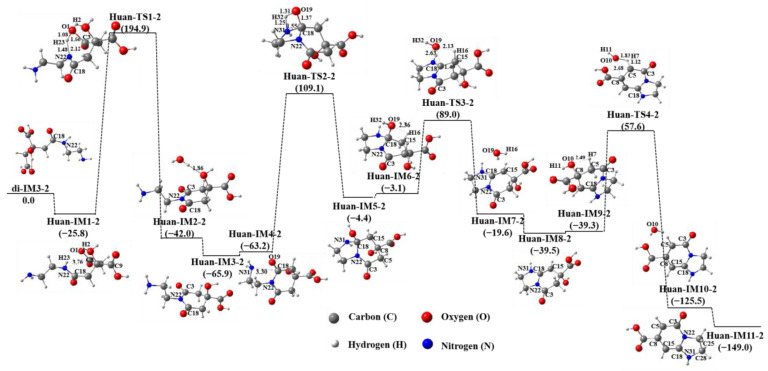
The first cyclization path of di-IM3-2 in an aqueous solution.

**Figure 5 molecules-29-02920-f005:**
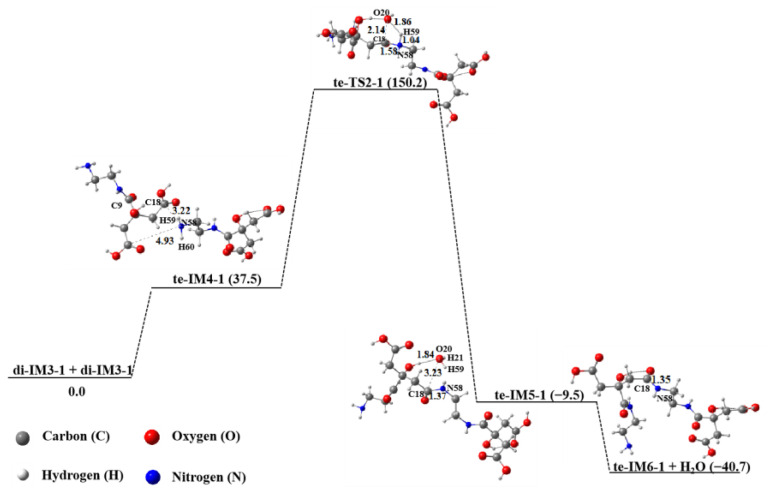
Polymerization of two di-IM3-1 molecules in aqueous solution.

**Figure 6 molecules-29-02920-f006:**
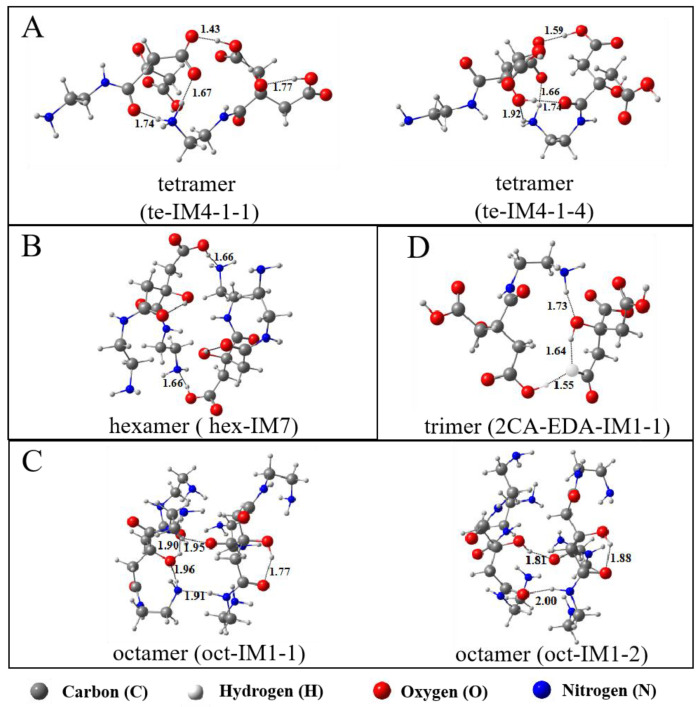
Optimized structure of oligomer products in aqueous solution produced from different molar ratio of EDA to CA: (**A**) 1.0, (**B**) 2.0, (**C**) 3.0, (**D**) 0.5.

**Figure 7 molecules-29-02920-f007:**
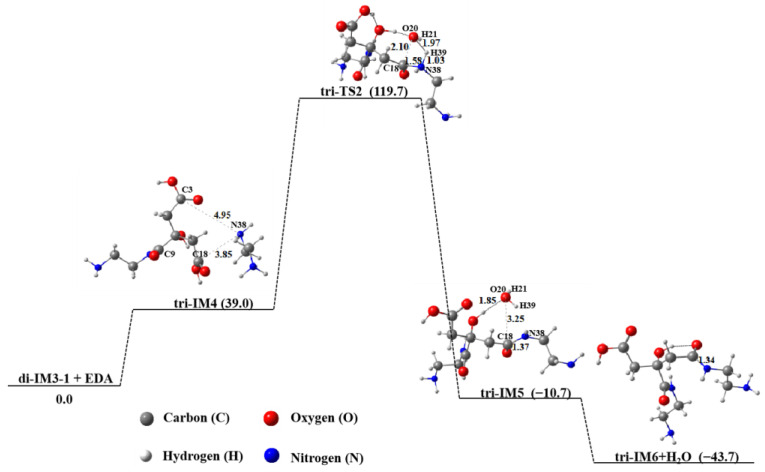
Polymerization of di-IM3-1 and EDA in aqueous solution.

**Figure 8 molecules-29-02920-f008:**
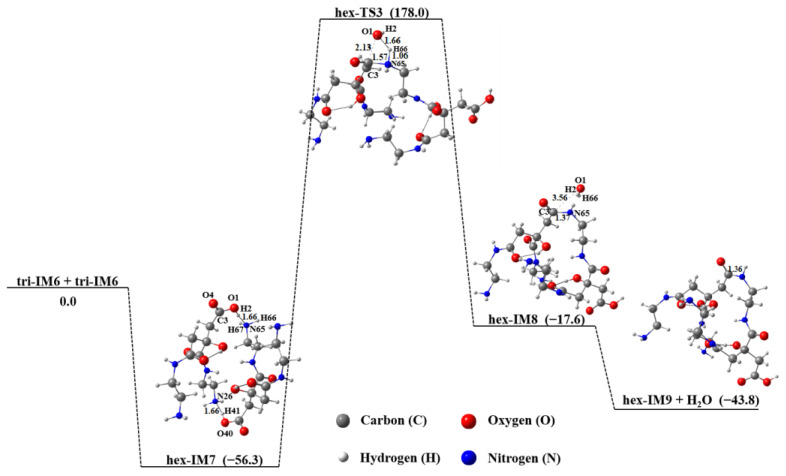
Polymerization of two tri-IM6 molecules in aqueous solution.

**Figure 9 molecules-29-02920-f009:**
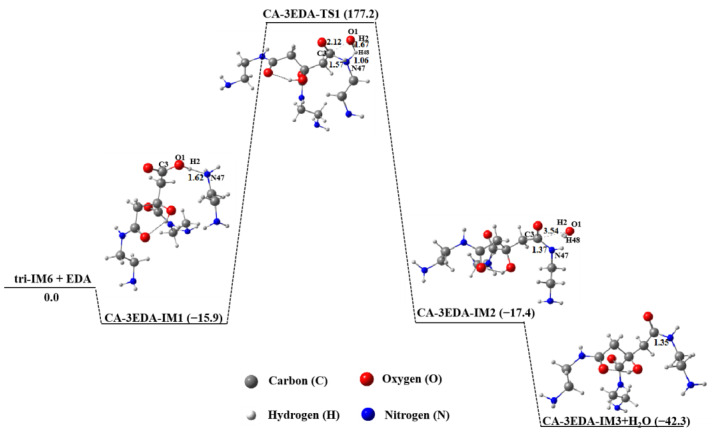
Polymerization of tri-IM6 and EDA in aqueous solution.

**Figure 10 molecules-29-02920-f010:**
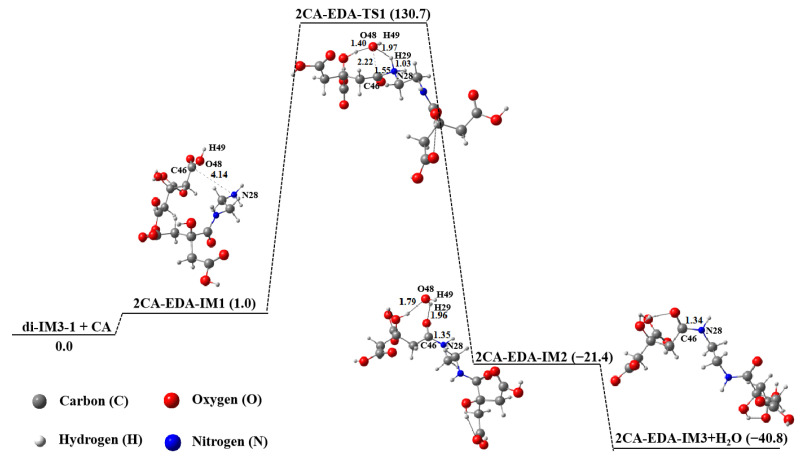
Polymerization of di-IM3-1 and CA in aqueous solution.

**Table 1 molecules-29-02920-t001:** The lowest frequency and Gibbs free energy of oligomeric isomers obtained under the weak interaction of two di-IM3-1 molecules in an aqueous solution.

Isomer	Frequency/cm^−1^	Gibbs Free Energy/Hartree	*^a^* ΔG/kJ∙mol^−1^
te-IM4-1-1	18	−1748.653039	0
te-IM4-1-2	20	−1748.637239	41.5
te-IM4-1-3	13	−1748.650431	6.9
te-IM4-1-4	19	−1748.652169	2.3
te-IM4-1-5	12	−1748.633345	51.7
te-IM4-1-6	17	−1748.644994	21.1
te-IM4-1-7	12	−1748.648700	11.4
te-IM4-1-8	21	−1748.626855	68.8
te-IM4-1-9	12	−1748.643023	26.3
te-IM4-1-10	18	−1748.633756	50.7
te-IM6-1 + H_2_O	8, 1610	−1748.649504	9.3

*^a^* ΔG is the difference in free energy of “te-IM6-1 + H_2_O” and other isomers relative to “te-IM4-1-1”.

**Table 2 molecules-29-02920-t002:** Lowest frequency and Gibbs free energy of CA-3EDA-IM3 and their oligomeric isomers formed under their weak interaction in aqueous solution.

Isomer	Frequency/cm^−1^	Gibbs Free Energy/Hartree	*^a^* ΔG/kJ∙mol^−1^
CA-3EDA-IM3 + CA-3EDA-IM3	12, 12	2204.917002	0
oct-IM1-1	10	−2204.922257	−13.8
oct-IM1-2	19	−2204.921466	−11.7
oct-IM1-3	17	−2204.910057	18.2
oct-IM1-4	14	−2204.917378	−1.0
oct-IM1-5	18	−2204.908480	22.4
oct-IM1-6	14	−2204.914584	6.4
oct-IM1-7	11	−2204.916880	0.3
oct-IM1-8	15	−2204.909931	18.6
oct-IM1-9	14	−2204.912994	10.5
oct-IM1-10	8	−2204.919530	−6.6

*^a^* ΔG is the difference in free energy of each oligomer isomer relative to “CA-3EDA-IM3 + CA-3EDA-IM3”.

**Table 3 molecules-29-02920-t003:** The lowest frequency and Gibbs free energy of oligomeric isomers formed under the weak interaction of di-IM3-1 and CA in an aqueous solution.

Isomer	Frequency/cm^−1^	Gibbs Free Energy/Hartree	*^a^* ΔG/kJ∙mol^−1^
2CA-EDA-IM1-1	13	−1634.583669	0
2CA-EDA-IM1-2	16	−1634.576321	19.3
2CA-EDA-IM1-3	15	−1634.569386	37.5
2CA-EDA-IM1-4	13	−1634.572770	28.6
2CA-EDA-IM1-5	46	−1634.562267	56.2
2CA-EDA-IM1-6	18	−1634.565983	46.5
2CA-EDA-IM1-7	9	−1634.578954	12.4
2CA-EDA-IM1-8	12	−1634.562583	55.4
2CA-EDA-IM1-9	19	−1634.564734	49.7
2CA-EDA-IM1-10	19	−1634.559758	62.8
2CA-EDA-IM3 + H_2_O	12, 1610	−1634.571953	30.8

*^a^* ΔG is the difference in free energy of “2CA-EDA-IM3 + H_2_O” and other isomers relative to “2CA-EDA-IM1-1”.

## Data Availability

The data presented in this study are available in the article and Appendix A.

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
