# Peer review of "A Theoretical Investigation into the Oligomer Structure of Carbon Dots Formed from Small-Molecule Precursors"

_molecules, 2024, doi:10.3390/molecules29122920_

Round 1

Reviewer 1 Report

Comments and Suggestions for Authors

In the manuscript “A Theoretical Investigation on the Oligomer Structure of Carbon Dots Formed from Small Molecule Precursors”, the authors present a nice investigation of the formation of carbon dots using citric acid and ethylenediamine molecules as precursors. They explore each precursor's molecular geometry and their preferred reaction sites according to their electrostatic potential. Then they showed the mechanism of dimerization, cyclization, and polymerization, discussing in detail the energy barrier and the resulting structure of each process.

I think there is a lot of good work behind the manuscript, the structure and presentation are generally good, and the presented results are relevant to the scientific community. Therefore, I recommend the manuscript be accepted for publication after a few minor issues that I would comment on in the following text are solved.

1.      In the first paragraph of the introduction, a better description of the top-down and bottom-up methods is necessary as the authors only mentioned briefly, providing insufficient reference to them.

2.      On page 2 line 53, while commenting on the work of Song et al. The authors mentioned that carbon dots are prepared with a “special method”. Can you please elaborate on what is special about that method, or better describe it in scientific terms.

3.      On page 2 line 66, the authors mentioned the experimental shortcomings of studying the carbon dots, but there is no reference to this. Please add the appropriate references in this section.

4.      I would prefer that the theoretical methods are presented before the results. I was following the order of the manuscript and got very confused when I started to read the results.

5.      On page 2 lines 84-85, the authors mentioned that, clearly the structure of CA and EDA is stable because neither of them has imaginary vibrational frequencies. But I can’t see the spectrum anywhere in the figure, so it is not as clear to me.

6.      On page 3 line 120, the authors say that C9 being the main reaction site of CA is consistent with previous results. But I can’t find any reference to those results.

7.      A question regarding the results of page 6, is why cyclization with C3 has a lower energy barrier than with C9, despite this being the preferable binding site in the dimerization?

8.      On page 8 line 253, the authors say that two tri-IM6 molecules are more easily connected by inter-molecular hydrogen bonds than by amidation reaction. But, does this not contradict the results presented in Figure 1 and discussed on page 3?

9.      As a general comment, the authors constantly mention that geometries are optimized in aqueous solution. However, the simulations are done using DFT, so molecules are isolated, in a vacuum. There are no additional atoms in the calculation to simulate the aqueous environment. Probably the optimized geometry is related to the ones encountered in the experiments under aqueous solution, but they are not optimized like that. So, please correct this statement that appears in several parts of the manuscript.

10.  On the conclusions the authors mention that their results have been proved from a thermodynamic perspective. Can you please elaborate on what you mean by a thermodynamic perspective? I agree with the results from an energetic perspective, but I don’t understand the assertion of the authors.

Comments on the Quality of English Language

No comments about the English quality

Author Response

Responses Letter (molecules-3044705)

First of all, we would like to thank reviewer for your valuable comments and suggestions, which are greatly helpful to us for further improvement of this paper. The following are the responses to the comments, point by point and the corresponding changes made in the manuscript are red marked.

Reviewer 1

Comments and Suggestions for Authors

In the manuscript “A Theoretical Investigation on the Oligomer Structure of Carbon Dots Formed from Small Molecule Precursors”, the authors present a nice investigation of the formation of carbon dots using citric acid and ethylenediamine molecules as precursors. They explore each precursor's molecular geometry and their preferred reaction sites according to their electrostatic potential. Then they showed the mechanism of dimerization, cyclization, and polymerization, discussing in detail the energy barrier and the resulting structure of each process.

I think there is a lot of good work behind the manuscript, the structure and presentation are generally good, and the presented results are relevant to the scientific community. Therefore, I recommend the manuscript be accepted for publication after a few minor issues that I would comment on in the following text are solved.

Author answers: Thank you very much for your recognition of our work. All of these improvements are due to your valuable comments and suggestions.

  1. In the first paragraph of the introduction, a better description of the top-down and bottom-up methods is necessary as the authors only mentioned briefly, providing insufficient reference to them.

Author answers: Sorry for the unspecific description. The descriptions of the “top-down” and “bottom-up” methods have been added in the revised manuscript. On page 1, two sentences “The“top-down” method mainly involves cutting large-sized carbon precursors (such as graphite, graphene, carbon nanotubes, carbon fibers, and carbon black) into small-sized carbon quantum dots using physical or chemical methods. The “bottom-up” methods involves using small molecules as precursors to obtain larger carbon quantum dots through a series of chemical reactions [7].” have been added in the revised manuscript.

  1. On page 2 line 53, while commenting on the work of Song et al. The authors mentioned that carbon dots are prepared with a “special method”. Can you please elaborate on what is special about that method, or better describe it in scientific terms.

Author answers: Sorry for the unspecific description. The separation method of carbon dots used in the work of Song et al. is column chromatography on silica. On page 2 line 61 in the revised manuscript, “special method” has been modified to “column chromatography on silica” in the revised manuscript.

  1. On page 2 line 66, the authors mentioned the experimental shortcomings of studying the carbon dots, but there is no reference to this. Please add the appropriate references in this section.

Author answers: Sorry for ignoring this issue. On page 2 line 66 (lines 75-76 in the revised manuscript), regarding the experimental shortcomings of studying carbon dots, reference [15] can reflect them. Conventional experimental methods may infer the presence of oligomers, but their specific structure cannot be obtained. In addition, a new reference “[19]” has been added to demonstrate that the oligomer structure of carbon dots is difficult to obtain through conventional methods.

On page 14, one reference “[19] Mintz, K. J.;  Bartoli, M.; Rovere, M.; Zhou, Y.-Q.; Hettiarachchi, S. D.; Paudyal, S.; Chen, J.-Y.; Domena, J. B.; Liyanage, P. Y.; Sampson, R.; Khadka, D.; Pandey, R. R.; Huang, S.-X.; Chusuei, C. C.; Tagliaferro, A.; Leblanc, R. M. A deep investigation into the structure of carbon dots. Carbon 2021,173, 433-447.” have been added as reference [19] in the revised manuscript.

  1. I would prefer that the theoretical methods are presented before the results. I was following the order of the manuscript and got very confused when I started to read the results.

Author answers: Thank you for your suggestion. On pages 2 and 3 in the revised manuscript, “Computational Methods” has been placed in front of the “Results and discussion”.

  1. On page 2 lines 84-85, the authors mentioned that, clearly the structure of CA and EDA is stable because neither of them has imaginary vibrational frequencies. But I can’t see the spectrum anywhere in the figure, so it is not as clear to me.

Author answers: Sorry for the unspecific description. We did not provide a vibration frequencies figure, but presented the data of the lowest vibration frequencies in Table S1(Supplementary Information). These data can indirectly indicate that our molecular structure is at a local minimum and is relatively stable.  

  1. On page 3 line 120, the authors say that C9 being the main reaction site of CA is consistent with previous results. But I can’t find any reference to those results.

Author answers: Sorry for the unspecific description. The meaning we want to express is that the calculation result "C9 preferentially reacts with EDA" is consistent with the ESP result "C9 is the main reactive site" mentioned above. Therefore, on page 4 line 159 in the revised manuscript, “previous ESP results” has been changed to “the above ESP results” in the revised manuscript.

  1. A question regarding the results of page 6, is why cyclization with C3 has a lower energy barrier than with C9, despite this being the preferable binding site in the dimerization?

Author answers: Thank you for your good question. The cyclization reaction process involves several steps. Although the energy barrier of the first step of cyclization for C9 is relatively low compared to C3, the energy barrier required for dehydrogenation of the cyclized product to form conjugated double bonds at the second cyclization path (133.3 kJ∙mol-1 and 241.3 kJ∙mol-1 respectively) is much higher than that of the first cyclization path (90.8 kJ∙mol-1 and 97.3 kJ∙mol-1 respectively), resulting in a high energy barrier for the entire reaction. Therefore, C3 is the preferable binding site in the cyclization process.

  1. On page 8 line 253, the authors say that two tri-IM6 molecules are more easily connected by inter-molecular hydrogen bonds than by amidation reaction. But, does this not contradict the results presented in Figure 1 and discussed on page 3?

Author answers: Thank you for your good question. tri-IM6 is obtained by the reaction of EDA and dimer product di-IM3-1. Both tri-IM6 and di-IM3-1 are connected by amidation reaction. However, for tri-IM6, when further aggregation occurs, due to the lower energy of the oligomer (hex-IM7) obtained through inter-molecular hydrogen bonds compared to its products connected through amide bonds (hex-IM9) , leading to the result that two tri-IM6 molecules are more easily connected by inter-molecular hydrogen bonds than by amidation reaction. Therefore, this result is not contradict the results presented in Figure 1 and discussed on page 3.

  1. As a general comment, the authors constantly mention that geometries are optimized in aqueous solution. However, the simulations are done using DFT, so molecules are isolated, in a vacuum. There are no additional atoms in the calculation to simulate the aqueous environment. Probably the optimized geometry is related to the ones encountered in the experiments under aqueous solution, but they are not optimized like that. So, please correct this statement that appears in several parts of the manuscript.

Author answers: Thank you for your good question. In the “Computational Methods” section of this revised manuscript, it is mentioned that “The geometric configuration optimization and frequency calculation of all species in this work were carried out under the consideration of the conductor-like polarizable continuum model (CPCM, water, ε = 78.4). In the preliminary study of the reaction process of small molecule compounds in aqueous solutions, CPCM approximation could be used to consider the impact of bulk water [25–28].” (On page 2 lines 91-96 in the revised manuscript). Therefore, the conductor-like polarizable continuum model (CPCM) is used to simulate water as a solvent during the calculations in this work.

  1. On the conclusions the authors mention that their results have been proved from a thermodynamic perspective. Can you please elaborate on what you mean by a thermodynamic perspective? I agree with the results from an energetic perspective, but I don’t understand the assertion of the authors.

Author answers: Thank you for your good question and suggestion. The term "thermodynamic perspective" has a wide range, while "energetic perspective" is relatively more specific and relevant. On page 12, one sentence “and the formation mechanism of small molecule IPCA had been proved from a thermodynamic perspective.” has changed to “and the formation mechanism of small molecule IPCA had been proved from a energetic perspective.” in the revised manuscript.

Reviewer 2 Report

Comments and Suggestions for Authors

With regard to the methodology presented in this work, it is necessary to conduct some modifications and a number of important verifications.

On the one hand, it is important to highlight that CPCM is not the optimal choice for modeling solvents in the manuscript and plenty of discrepancies can be observed when compared with experimental results. However, this work can be regarded as a preliminary step in understanding the nature of these kinds of systems. This is a positive point to consider this work but a warning must be included in the manuscript.

On the other hand, the authors present very low frequencies while using Gaussian09, whose default DFT grid is small, which results in unreliable results in those cases. In addition, It is important to note that the stability of the obtained wave functions has not been considered and must be verified.

More details and discussion are included in the attached PDF file.

Comments on the Quality of English Language

The manuscript is not well-written. Errors in spelling, punctuation, and grammar must be corrected to ensure its suitability for publication. A list of examples (which is not exhaustive) is included in the attached PDF file.

Author Response

Responses Letter (molecules-3044705)

First of all, we would like to thank reviewer for your valuable comments and suggestions, which are greatly helpful to us for further improvement of this paper. The following are the responses to the comments, point by point and the corresponding changes made in the manuscript are red marked.

Reviewer 2

Comments and Suggestions for Authors

With regard to the methodology presented in this work, it is necessary to conduct some modifications and a number of important verifications.

On the one hand, it is important to highlight that CPCM is not the optimal choice for modeling solvents in the manuscript and plenty of discrepancies can be observed when compared with experimental results. However, this work can be regarded as a preliminary step in understanding the nature of these kinds of systems. This is a positive point to consider this work but a warning must be included in the manuscript.

On the other hand, the authors present very low frequencies while using Gaussian09, whose default DFT grid is small, which results in unreliable results in those cases. In addition, It is important to note that the stability of the obtained wave functions has not been considered and must be verified.

More details and discussion are included in the attached PDF file.

The manuscript presented by C. Li, X. Zhu and M. Xu, entitled “A Theoretical Investigation on the Oligomer Structure of Carbon Dots Formed from Small Molecule Precursors,” examines the formation of carbon dots from citric acid (CA) and ethylenediamine (EDA). To conduct this study, the authors employed the transition state theory with a continuous solvent (CPCM with water) and explored various pathsways, including the dimerization of CA and EDA, cyclization of intermediates and polymerization processes.

The manuscript contains sufficient work to be a valuable contribution to the field as a publication in Molecules. Moreover, the topic and the aim of the work are of considerable interest. Obtaining all these intermediates and TSs are a complex task, but it appears to have been successfully accomplished.

 Nevertheless, I have identified several significant inconsistencies, which necessitate a MAJOR revision.

Author answers: Thank you very much for your valuable comments and suggestions.

  1. In general, the manuscript is not well-written. Errors in spelling, punctuation, and grammar must be corrected to ensure its suitability for publication. I have compiled a list of examples (which is not exhaustive):
  • The Latin expression "et al." is formed by two words. The authors employ the term "etal" incorrectly in a systematic manner.

Author answers: Sorry for these mistakes. All "etal" in the revised manuscript have been changed to "et al." and marked with a special red color (On page 2 in the revised manuscript).

  • It is necessary to revise the use of acronyms, as some of them are undefined.

Author answers: Sorry for these mistakes. All the acronyms of in the revised manuscript have been checked and revised, and marked with a special red color.

On page 1, “thus obtaining IPCA under certain conditions” has been changed to “IPCA (5-oxo-1,-2,3,5-tetrahydroimidazo [1,2-a]pyridine-7-carboxylic acid) could therefore be obtained under certain conditions.” in the revised manuscript.

On page 2, “The precise structure of IPCA was proved by FTIR, NMR and LC-MS techniques” has been changed to “The precise structure of IPCA was proved by FTIR, NMR and liquid chromatography-mass spectrometry (LC-MS) techniques” in the revised manuscript.

On page 2, “The structure of simple compounds can be proven through conventional analytical techniques, such as FTIR, NMR, XPS” has been changed to “The structure of simple compounds can be proven through conventional analytical techniques, such as FTIR, NMR, and X-ray Photoelectron Spectroscopy (XPS)” in the revised manuscript.

  • It is advisable to avoid the use of very long and complex sentences. In several cases, the authors failed to maintain concordance between the core of the subjects and the main verb. For example:

â—¦ Line 73: […] and the reaction process of CA and EDA […] *were […].

Author answers: Sorry for this mistake. On page 2 lines 81-84 in the revised manuscript, “The electrostatic potential (ESP) of CA had been evaluated by Multiwfn to analyze its surface reaction sites, and the reaction process of CA and EDA in different molar ratios to obtain oligomers were in-depth investigated by DFT method for the first time.” has been changed to “The electrostatic potential (ESP) of CA was evaluated by Multiwfn to analyze its surface reaction sites. The reaction process of CA and EDA in different molar ratios to obtain oligomers was investigated in depth using the DFT method for the first time.”.

â—¦ In Line 354: “the oligomers of CDs obtained from different molar ratios of EDA and CA *was revealed by DFT-D3 method.” By the way, a method reveals or the conclusions of the results reveal?

Author answers: Sorry for this mistake. The meaning we want to express is that the structure of oligomers was obtained by using DFT-D3 method.

  On page 12 lines 376-377 in the revised manuscript, “In summary, the oligomers of CDs obtained from different molar ratios of EDA and CA was revealed by DFT-D3 method,” has been changed to “In summary, the oligomers of CDs at different molar ratios of EDA and CA were obtained by using the DFT-D3 method.”.

  • Line 63: “[…] as FTIR, NMR, XPS” → “and XPS”.

Author answers: Sorry for this mistake. On page 2 lines 71-73 in the revised manuscript, “The structure of simple compounds can be proven through conventional analytical techniques, such as FTIR, NMR, XPS” has been changed to “The structure of simple compounds can be proven through conventional analytical techniques, such as FTIR, NMR, and X-ray Photoelectron Spectroscopy (XPS)”.

  • Line 177-178: “[…] reported in the literature15.” → reported in the literature [15].

Author answers: Sorry for this mistake. On page 6 lines 218-220 in the revised manuscript, “The structure of Huan-IM11-2 is consistent with that of IPCA (5-oxo-1,-2,3,5-tetrahydroimidazo [1,2-a]pyridine-7-carboxylic acid) reported in the literature 15” has been changed to “The structure of Huan-IM11-2 is consistent with that of IPCA (5-oxo-1,-2,3,5-tetrahydroimidazo [1,2-a]pyridine-7-carboxylic acid) reported in the literature [15]”.

  • Line 197: “Figure 5 shows the *linerpolymerization […]” → “linear”

Author answers: Sorry for this mistake. On page 6 lines 241-242 in the revised manuscript, “Figure 5 shows the liner polymerization process of two di-IM3-1 molecules in aqueous solution” has been changed to “Figure 5 shows the linear polymerization process of two di-IM3-1 molecules in aqueous solution”.

  • Line 199: “First, the N58 atom of amino group in one di-IM3-1 molecule closes to the C18 atom of carboxyl in another di-IM3-1 molecule, forming the intermediate te-IM4-1.” → “First, the N58 atom of the amino group in one di-IM3-1 molecule close to the C18 atom of the carboxyl in another di-IM3-1 molecule, forming the intermediate te-IM4-1.”

Author answers: Sorry for this mistake. On page 6 lines 243-245 in the revised manuscript, “First, the N58 atom of amino group in one di-IM3-1 molecule closes to the C18 atom of carboxyl in another di-IM3-1 molecule, forming the intermediate te-IM4-1” has been changed to “First, the N58 atom of the amino group in one di-IM3-1 molecule close to the C18 atom of the carboxyl in another di-IM3-1 molecule, forming the intermediate te-IM4-1”.

  • Line 212: “This result indicates that two di-IM3-1 molecules are prone to spontaneously form te-IM4-1-1 through inter-molecular interaction rather than form te-IM6-1 by amidation reaction.” → “This result suggests that two di-IM3-1 molecules are more likely to spontaneously form te-IM4-1-1 through intermolecular interaction rather than forming te-IM6-1 through an amidation reaction.”

Author answers: Sorry for this mistake. On page 7 lines 256-258 in the revised manuscript, “This result indicates that two di-IM3-1 molecules are prone to spontaneously form te-IM4-1-1 through inter-molecular interaction rather than form te-IM6-1 by amidation reaction” has been changed to “This result suggests that two di-IM3-1 molecules are more likely to spontaneously form te-IM4-1-1 through intermolecular interaction rather than forming te-IM6-1 through an amidation reaction”.

  • Line 236: a very long sentence (> 3 lines). In addition, it includes some mistakes: “Although di-IM3-1 molecules are easy to form tetramers by inter-molecular hydrogen bonds, they tend to polymerize with small molecules EDA to obtain the product tri-IM6 due to the presence of excessive EDA when the molar ratio of EDA to CA is 2.0, as depicted in Figure 7.” →“Although di-IM3-1 molecules are readily formed into tetramers through intermolecular hydrogen bonds, they tend to polymerize with small molecules of EDA to yield the product tri-IM6. This is due to the presence of excessive EDA when the molar ratio of EDA to CA is 2.0, as depicted in Figure 7.”

Author answers: Sorry for these mistakes. On page 8 lines 281-284 in the revised manuscript, “Although di-IM3-1 molecules are easy to form tetramers by inter-molecular hydrogen bonds, they tend to polymerize with small molecules EDA to obtain the product tri-IM6 due to the presence of excessive EDA when the molar ratio of EDA to CA is 2.0, as depicted in Figure 7” has been changed to “Although di-IM3-1 molecules are readily formed into tetramers through intermolecular hydrogen bonds, they tend to polymerize with small molecules of EDA to yield the product tri-IM6. This is due to the presence of excessive EDA when the molar ratio of EDA to CA is 2.0, as depicted in Figure 7”.

  • It is advisable to refrain from the overuse of the terms "form" and derivatives. The authors employed this verb with considerable frequency in the text, which resulted in a notable decline in the quality of the text. For instance, the paragraph in line 313 contains the terms "formation," "form," and "forms" on five consecutive lines.

Author answers: Sorry for these mistakes. We have replaced the word "form" in many sentences with other words. The specific modification details are marked in red in the revised manuscript.

  • Line 335: “zero imaginary frequency” → “zero imaginary frequencies”

Author answers: Sorry for these mistakes. On page 3 line 100 in the revised manuscript, “zero imaginary frequency” has been changed to “zero imaginary frequencies”.

  • After semicolon the first letter is not in uppercase.

Author answers: Sorry for these mistakes.

On page 6 line 213 in the revised manuscript, “While the free energy of Huan-IM11-1 only decreases by 113.2 kJ∙mol−1,” has been changed to “while the free energy of Huan-IM11-1 only decreases by 111.7 kJ∙mol−1,”.

On page 11 lines 337-338 in the revised manuscript, “Protons in the carboxyl groups of oct-IM1-1 and oct-IM1-2 will not ionize and migrate,” has been changed to “protons in the carboxyl groups of oct-IM1-1 and oct-IM1-2 will not ionize and migrate,”.

On page 11 lines 364-365 in the revised manuscript, “Excessive CA does not participate in the amidation reaction,” has been changed to “excessive CA does not participate in the amidation reaction,”.

On page 12 lines 401-403 in the revised manuscript, “Lowest frequency and Gibbs free energy of CA, EDA, and some oligomeric isomers formed by weak interaction; Dimerization of CA in aqueous solution; The second cyclization path of di-IM3-2 in aqueous solution.” has been changed to “Lowest frequency and Gibbs free energy of CA, EDA, and some oligomeric isomers formed by weak interaction; dimerization of CA in aqueous solution; the second cyclization path of di-IM3-2 in aqueous solution.”.

Therefore, it is recommended that the manuscript be subjected to a general revision of its writing.

Author answers: Thank you again for pointing out these mistakes. The writing of this manuscript has been revised by the professional English editing department of MDPI (The changes made in the revised manuscript are marked in purple).

  1. With regard to the methodology presented in this work, it is necessary to make a number of important observations.
  • The use of a continuous solvent model, such as CPCM, in particular when employing a polar solvent such as water, is not a wholly optimal choice. It would be preferable to utilize an adequate and more appropriate methodology, such as QM/MM or Molecular Dynamics calculations, to achieve a more precise result. However, this work can be regarded as a preliminary step in understanding the nature of these kinds of systems and the associated reactions. The authors must highlight this aspect within the text. Furthermore, it would be beneficial to revisit the distinctions between other solvent models, such as CPCM and SMD, although this may be more appropriately addressed in a separate publication.

Author answers: Thank you for your comments and suggestions. We have emphasized in the revised manuscript that, in the preliminary study of the reaction process of small molecule compounds in aqueous solutions, CPCM approximation could be used to consider the impact of bulk water.

On page 2, “The geometric configuration optimization and frequency calculation of all species in this work were carried out under the consideration of the conductor-like polarizable continuum model (CPCM, water, ε = 78.4)[25–28]” has been changed to “The geometric configuration optimization and frequency calculation of all species in this work were carried out under the consideration of the conductor-like polarizable continuum model (CPCM, water, ε = 78.4). In the preliminary study of the reaction process of small molecule compounds in aqueous solutions, CPCM approximation could be used to consider the impact of bulk water [25–28]”.

Moreover, we have calculated a set of structures in this work using SMD as an implicit solvent model. Compared to the CPCM implicit solvent model, the Gibbs free energy calculated using SMD is slightly lower. However, the conclusions obtained by using SMD as the implicit solvent model for our work are consistent with those acquired by using CPCM, indicating that the CPCM model is feasible as a preliminary exploration of the reaction process of CA and EDA in aqueous solutions. As shown in Tables S2.

Table S2. Lowest frequency and Gibbs free energy of dimeric isomers formed under the weak interaction of CA and EDA in aqueous solution

Isomer

1Frequency

/cm-1

1Gibbs free energy/hartree

1, a DG 

/kJ∙mol-1

2Frequency

/cm-1

2Gibbs free energy/hartree

2, a DG 

/kJ∙mol-1

di-IM1-1-1

13

-950.774884

15.0

16

-950.799931

18.2

di-IM1-1-2

29

-950.759430

55.6

21

-950.783220

60.5

di-IM1-1-3

14

-950.774570

15.8

22

-950.797666

22.5

di-IM1-1-4

6

-950.774590

15.8

15

-950.796895

24.6

di-IM1-1-5

12

-950.773911

17.5

21

-950.796897

24.6

di-IM1-1-6

23

-950.765761

38.9

17

-950.794159

31.7

di-IM1-1-7

16

-950.773357

19.0

19

-950.796945

24.4

di-IM1-1-8

33

-950.764401

42.5

23

-950.794266

31.5

di-IM1-1-9

22

-950.764419

42.5

20

-950.793416

33.7

di-IM1-1-10

19

-950.765895

38.6

31

-950.793112

34.5

di-IM3-1+H2O

21, 1610

-950.780592

0

29, 1589

-950.806244

0

1 Lowest frequency and Gibbs free energy of dimeric isomers were carried out under the consideration of the conductor-like polarizable continuum model (CPCM).

2 Lowest frequency and Gibbs free energy of dimeric isomers were carried out under the consideration of solvation model based on density (SMD).

a DG is the difference in free energy of each isomer relative to “di-IM3-1 + H2O”.

Comparison results: Under the same conditions in the computing environment, the Gibbs free energy obtained using the SMD solvent model is slightly lower than that acquired using the CPCM solvent model, but the decreasing trend of their free energy is basically the same. The conclusions obtained from the two implicit solvent models are consistent: the Gibbs free energy of the dimer product (di-IM3-1 + H2O) connected by amide bonds is lower than that of other dimer isomers connected by intermolecular hydrogen bonding. CA and EDA will preferentially combine through amide bonds to obtain the product di-IM3-1.

  • The authors do not clearly describe the concept of stability and minimum. For example, in Line 84 the authors state “[…] the optimized structure of CA and EDA is stable, because neither of these structures has imaginary vibrational frequency.” The structure is simply a minimum according to the frequencies calculations. To verify the stability of the wavefunction, one must carry out a calculation with the option “stable” in Gaussian (https://gaussian.com/stable/): “Note that analytic frequency calculations are only valid if the wavefunction has no internal instabilities. In examining the results prior to a frequency calculation, it suffices to see if any singlet instabilities exist for restricted wavefunctions or if any instabilities (singlet or triplet) exist for unrestricted wavefunctions.”

This is also remarked in the section about frequencies calculations in the Gaussian manual (https://gaussian.com/freq/):

“Note also that the CPHF (coupled perturbed SCF) method used in determining analytic frequencies is not physically meaningful if a lower energy wavefunction of the same spin multiplicity exists. Use the Stable keyword to test the stability of Hartree-Fock and DFT wavefunctions.”

Author answers: Sorry for the unspecific description. 

On page 3 lines 99-101 in the revised manuscript, “From the calculated vibration frequencies, it could be judged whether the optimized structure was stable (zero imaginary frequency) and whether there was only one imaginary frequency in the transition state.” has been changed to “The calculated vibration frequencies could be used to judged whether the optimized structure was a local minimum point (zero imaginary frequencies) and whether there was only one imaginary frequency in the transition state.”.

On page 3 lines 123-125 in the revised manuscript, “Clearly, the optimized structure of CA and EDA is stable, because neither of these structures has imaginary vibrational frequency.” has been changed to “Clearly, neither of these structures has imaginary vibrational frequency, indicating that the structure of CA and EDA is a local minimum point.”.

Moreover, the authors must ascertain the stability of all the wavefunctions obtained through this kind of Gaussian calculation to ensure the reliability of the stationary points obtained. The computational cost is less than that of a frequency calculation.• With the very low frequencies obtained by the authors, all CPKS calculations must be repeated with the option “integral=ultrafinegrid” to obtain reliable frequencies. According to the Gaussian manuals:

  • With low frequencies obtained by the authors, it is imperative that all CPKS calculations be repeated with the "integral=ultrafinegrid" option in order to ensure the reliability of the obtained frequencies. In accordance with the instructions provided in the Gaussian manuals, this is highly recommended:

“[…] UltraFine requests a pruned (99,590) grid. It is recommended for molecules containing lots of tetrahedral centers and for computing very low frequency modes of systems. This grid is also useful for optimizations of larger molecules with many soft modes such as methyl rotations, making such optimizations more reliable. […]

The default grid is “UltraFine” in Gaussian 16. In contrast, the default grid in all versions of Gaussian 09 is “FineGrid.” (https://gaussian.com/g16/g09ur.tgz). However, the G09 manual also recommends the use of “integral=ultrafinegrid” with these kinds of situations.

The authors are unable to confirm that the structures obtained in this work are minima or TSs without a reliable DFT grid.

Author answers: Thanks for your good suggestions. We have recalculated all structures in this work by adding the integral=ultrafinegrid keyword. The data has been updated in the revised manuscript and marked in red. Compared with the data obtained without using the integral=ultrafinegrid keyword, the changes in each group of data are minimal and have no impact on our previous analysis conclusions.

   On page 2 lines 97-98, one sentence “The integral=ultrafinegrid keyword was used simultaneously to ensure the reliability of the obtained frequencies.” has been added in the revised manuscript. 

  1. Some more questions to improve in the main text:
  • Ref. 7 seems to be an application to add in Line 31 together with References 1-6.

Author answers: Thank you for your reminder. Reference [7] reviewed the applications of carbon dots, but also summarized the methods for synthesizing carbon dots, including the "top-down" and "bottom-up" methods. On page 1 line 32 in the revised manuscript, “low cytotoxicity and good biocompatibility [1–6]” has been changed to “low cytotoxicity and good biocompatibility [1–7]”.

  • In line 39, I wonder the authors mean about the relationship between structure and performance. They should develop the idea with one sentence more to be clear.

Author answers: Thank you for your good suggestion. On pages 1-2 lines 44-48 in the revised manuscript, “The relationship between the structure and performance of doped CDs is still unclear, which leads to its limitation in many applications.” has been changed to “It is well-known that the structure of a material determines its performance, and performance determines its function and application. The structure of doped CDs is still unclear, and the relationship between their structure and performance cannot be established, which leads to limitation in many applications.”.

Round 2

Reviewer 2 Report

Comments and Suggestions for Authors

I would like to thank the authors for all their hard work in preparing the second version of their manuscript. I think it is publishable in the present form.